# Hybrids Composed of an Fe-Containing Wells–Dawson Polyoxometalate and Carbon Nanomaterials as Promising Electrocatalysts for the Oxygen Reduction Reaction

**Hugo C. Novais [1], Bruno Jarrais [1], Israël-Martyr Mbomekallé [2], Anne-Lucie Teillout [2], Pedro de Oliveira [2], Cristina Freire [1] and Diana M. Fernandes [1,\*]**

[1] LAQV-REQUIMTE/Department of Chemistry and Biochemistry, Faculty of Sciences, University of Porto, 4169-007 Porto, Portugal; hugodcnt@gmail.com (H.C.N.); bjarrais@gmail.com (B.J.); acfreire@fc.up.pt (C.F.)

[2] Equipe d'Electrochimie et de Photo-Électrochimie, Institut de Chimie Physique, UMR 8000 CNRS, Faculté des Sciences d'Orsay, Université Paris-Saclay, F-91405 Orsay, France; israel.mbomekalle@universite-paris-saclay.fr (I.-M.M.); anne-lucie.teillout@universite-paris-saclay.fr (A.-L.T.); pedro.almeida-de-oliveira@universite-paris-saclay.fr (P.d.O.)

\* Correspondence: diana.fernandes@fc.up.pt

**Abstract:** The oxygen reduction reaction (ORR) is a key cathodic reaction in energy-converting systems, such as fuel cells (FCs). Thus, it is of utmost importance to develop cost-effective and efficient electrocatalysts (ECs) without noble metals to substitute the Pt-based ones. This study focuses on polyoxometalate (POM)-based ECs for ORR applications. A Wells–Dawson POM salt $K_7$ $[P_2W_{17}(FeOH_2)O_{61}]\cdot 20H_2O$ was immobilised onto graphene flakes and multiwalled carbon nanotubes doped with nitrogen, denominated as $P_2W_{17}Fe@GF\_N8$ and $P_2W_{17}Fe@MWCNT\_N8$. The successful preparation of the composites was proved with various characterisation techniques, including FTIR, XPS and SEM. Both materials showed good ORR performance in an alkaline medium with similar potential onset values of ~0.84 V vs. RHE and diffusion-limiting current densities of $-3.9$ and $-3.3$ mA cm$^{-2}$ for $P_2W_{17}Fe@MWCNT\_N8$ and $P_2W_{17}Fe@GF\_N8$, respectively. Furthermore, both composites presented low Tafel slopes (48–58 mV dec$^{-1}$). Chronoamperometric tests revealed that the as-prepared nanocomposites rendered a significant improvement achieving between 90 and 94% of current retention in tolerance to methanol in comparison with Pt/C, and moderate to good long-term electrochemical stability with current retentions comprised between 68 and 88%. This work reinforces the use of POMs as important electroactive species for the preparation of alternative ORR electrocatalysts, exhibiting good activity, stability and selectivity towards the ORR in the presence of methanol.

**Keywords:** oxygen reduction reaction; electrocatalysis; carbon-based electrocatalyst; polyoxometalates; N-doping

## 1. Introduction

The population growth has resulted in a steep rise in energy demand, which has raised concerns about the security of our energy supplies in the midterm future. It is, therefore, of the highest urgency to develop sustainable alternative energy sources and conversion devices in order to reduce carbon emissions [1]. Fuel cells (FCs) present themselves as one of the most promising green options and sustainable technologies available. A key feature of these systems is the oxygen reduction reaction (ORR) that occurs at the cathode and generally exhibits slow kinetics [2–4]. Electrocatalysts (ECs) play a major role in most energy conversion devices, including FCs, because they enhance the selectivity and efficiency of the chemical reactions involved. Thus far, the most effective ECs for ORR, to date, have been derived from noble metals, like platinum and palladium [5]. The latter have several drawbacks, including expensive price, limited availability and a lack of

tolerance towards fuel crossover (such as methanol and CO) [5]. Since the discovery of the ORR in fuel cells, researchers have been working to improve the efficiency and stability of the electrocatalysts used in this process. In recent years, there have been significant advances in the development of ORR electrocatalysts, particularly in the use of iron-based catalysts [6]. Additionally, surface science studies have been conducted to understand better the electronic structure, stability and electrocatalytic properties of various catalysts [7,8]. In addition, computational methods have been developed to aid in the design of solid catalysts, which has opened up new possibilities for ORR electrocatalyst research [9]. Over recent years, several studies have been carried out with the objective of designing high-performance ECs based on earth-abundant elements to overcome the referred limitations associated with Pt- and Pd-based electrocatalysts [2,4,10]. The development of efficient and cost-effective catalysts for the ORR is of critical importance for the widespread adoption of fuel cells. Plus, significant progress has been achieved towards the design of ORR catalysts based on earth-abundant elements [11]. However, despite these advances, the search for efficient and stable ORR catalysts remains a topic of active research. One of the first catalysts tested for the ORR was cobalt phthalocyanine (CoPc) [11]. Jasinski's seminal work published in *Nature* in 1964 demonstrated that CoPc can catalyse ORR in acidic media. This work laid the foundation for the development of metal–phthalocyanine-based catalysts for fuel cells. Currently, the FCs that have received the most attention are proton exchange membrane fuel cells (PEMFCs) [12]. Although these systems are considered the most technologically advanced, they depend on pricey acid membranes and ionomers [13] and on platinum group metal-based electrocatalysts, which are costly, show limited durability and are scarcely available [14]. In order to enhance the electrocatalytic behaviour of the ORR, Pt and Pd have been incorporated with several transition metals, such as Ru [15], Ni [16], Ag [16] or Co [17,18].

Carbon nanomaterials (CMs), including graphene (GF) and carbon nanotubes (CNT), show great promise as efficient ORR electrocatalysts due to their high stability, conductivity, low cost and large surface area [19]. These materials demonstrate an exceptional high chemical and thermal stability, favourable electrical properties, high porosity and a large surface area [20]. Nanotubes and graphene are among the most widely applied materials in the ORR, employed as metal-free ECs in alkaline media and supporting materials for electrocatalysts in hybrid nanocomposites [21,22]. In addition, doping the material with heteroatoms, such as nitrogen, sulphur and phosphorus, could enhance its electrocatalytic activity [21–26]. Regarding nitrogen species, different forms can be incorporated into the graphitic network, including pyridinic, pyrrolic, graphitic (quaternary) and oxidized nitrogen, and these variations can significantly affect the electrocatalytic performance of the material. Artyushkova et al. showed that pyridinic and graphitic (quaternary) nitrogen species can act as highly active sites for the ORR due to their ability to stabilize the adsorption of reaction intermediates [26]. On the other hand, pyrrolic and oxidized nitrogen species have been found to exhibit lower activity for the ORR. Therefore, it is important to carefully consider the type and concentration of nitrogen species when designing doped carbon catalysts for the ORR. Incorporating nitrogen in the $sp^2$ carbon allows for the improvement of the electronic properties of pristine carbons, which gives rise to an enhanced electrocatalytic performance towards the ORR [21,22,27–29]. Over the years, several studies have reported progress in the performance of pyrolyzed catalysts for the ORR [30,31]. In 2009, Lefèvre et al. disclosed the utilisation of polymers as carbon–nitrogen sources for the synthesis of ORR catalysts [32]. The sacrificial support method (SSM) has also been shown to be an effective strategy for the preparation of ORR catalysts [33]. In 2011, Zelenay reported on a significant improvement in the durability of ORR catalysts based on nonprecious metals [34].

Furthermore, it has been proposed that creating composite electrocatalysts, which combine the exceptional properties of carbon nanomaterials with the vast stability and abundant redox characteristics of polyoxometalates (POMs), could be a feasible approach to increase the proficiency of various energy-linked reactions [28,35–38]. The POM used

in this work belongs to the Wells–Dawson family ($[(P_2W_{17}O_{61})Fe^{III}(H_2O)]^{7-}$). The iron-substituted POMs are of particular interest, as they may act either as oxidation or as reduction electrocatalysts [12]. Fe-substituted polyoxometalates (Fe-POMs) have emerged as a promising candidate amidst various catalysts due to their distinctive characteristics. These offer a cost-effective alternative to expensive Pt-based catalysts, making them a potential substitute for the ORR [39]. Moreover, the use of Fe in these catalysts adds to their sustainability. Iron is one of the most abundant elements on Earth, making it a readily available resource. It is also widely recycled, and its extraction and processing require less energy compared to other metals like cobalt [40].

This report details the preparation of composites by incorporating $K_7 [P_2W_{17}(FeOH_2)O_{61}]$. $\cdot 20H_2O$, a salt of Wells-Danson POM, onto two distinct nitrogen-doped carbon materials (nitrogen-doped graphene flakes—GF_N8; nitrogen-doped multiwalled carbon nanotubes—MWCNT_N8. The resulting composites were then applied as electrocatalysts for the ORR.

## 2. Experimental Section

*Materials Preparation and Characterisation Methods*

The preparation of the composites and ORR studies' solvents and materials are extensively described in the Supplementary Information (SI). Nitrogen doping of the carbon materials (0.60 g) was achieved with the mechanical ball-milling treatment during 5 h at a constant frequency of 15 vibrations $s^{-1}$ with Retsch MM200 equipment using melamine (0.26 g) as a precursor. Then, the materials that resulted underwent a thermal treatment under a $N_2$ flow (100 $cm^3$ $min^{-1}$) at a rate of 10 °C $min^{-1}$ until reaching the final temperature of 800 °C and kept at these temperature during 1 h; they were then cooled to room temperature under a nitrogen atmosphere and stored in a desiccator [41]. The $P_2W_{17}Fe@CM$ composites were then prepared through the immobilization of $P_2W_{17}Fe$ onto previously prepared MWCNT_N8 and GF_N8. The immobilization of the POM was achieved as follows: a 5 mL acetate buffer solution (pH = 4.0) containing 50 mg of $P_2W_{17}Fe$ was added to a 20 mL acetate buffer solution (pH = 4.0) containing 25 mg of CM. The mixture was dispersed for 15 min in an ultrasonic bath and then left to stir for 2 h at 400 rpm. Then, the resulting composites were filtered, washed and left to dry at 60 °C under vacuum overnight. The composites were labelled as $P_2W_{17}Fe@GF\_N8$ and $P_2W_{17}Fe@MWCNT\_N8$.

Before use in the electrocatalytic studies, the composites prepared were characterized using different techniques: Fourier-transformed infrared spectroscopy (FTIR), scanning electron microscopy (SEM), energy-dispersive X-ray spectroscopy (EDS) and X-ray photoelectron spectroscopy (XPS). The methods and equipment used are detailed in the Supplementary Materials. Electrocatalytic performances towards the ORR were evaluated using a PGSTAT 302N potentiostat (Metrohm Autolab B.V., Utrecht, The Netherlands) controlled with NOVA 2.1 software and using a conventional 3-electrode system. For all details regarding electrodes, electrode conditioning and modification, see the Supplementary Materials.

## 3. Results and Discussion

*3.1. Materials Characterisation*

All the prepared materials were initially characterized with infrared spectroscopy (Figure 1). The FTIR spectra of the nitrogen-doped carbon materials are shown in Figure S1. The MWCNT_N8 spectrum revealed several vibrational bands at 3436, 2928, 2846, 1641, 1383 and 1124 $cm^{-1}$, but some of these bands appeared quite weak and poorly resolved. This may have arisen from the low loading of the POMs onto the carbon material or their interaction with the graphene and carbon nanotube matrix that hindered their identification. The vibrational bands of the OH groups could be observed at 3436 $cm^{-1}$, and were due to the O-H stretching and bending modes from the adsorbed residual water and hydroxyl groups [42,43]. The C-N stretching vibration band was shown at 1383 $cm^{-1}$ [44,45]. The band at 1124 $cm^{-1}$ was assigned to the phenolic groups' vibration modes [42,46,47]. Lastly, the two bands at 2928 and 2846 $cm^{-1}$ corresponded to the C-H stretching modes, possibly caused by defects in the $sp^2$ hybridised domains. The band located at 1641 $cm^{-1}$ could be

attributed to the C=N and C=O stretching modes of carboxylic acid, carbonyl, ketone and quinone groups [42,44–47]. The vibrational spectra of GF_N8 were similar (Figure S1, black line), showing vibrational bands assigned to O-H stretching and bending modes at ~3410 and 1386 cm$^{-1}$, respectively [42,43]. The observed vibrations at 1630 cm$^{-1}$ represented the C=N and C=O stretching modes, while the phenol groups displayed vibrations at 1387 cm$^{-1}$ and the C-O stretching modes of oxygen groups could be observed at 1142 cm$^{-1}$ [38,42–45]. Finally, the band associated with the C-O-C and the C-O stretching modes and the C-O-H bending mode was observed at 1070 cm$^{-1}$ [42,46,47].

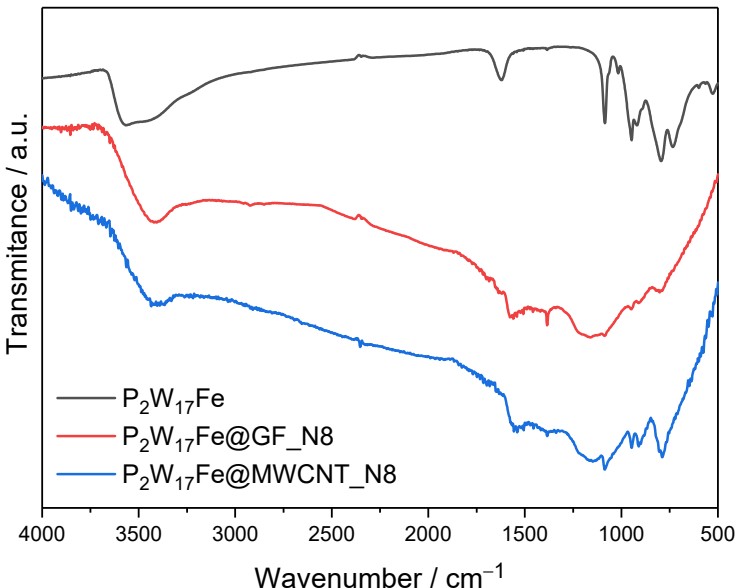

**Figure 1.** FTIR spectra in the 4000–500 cm$^{-1}$ range of K$_7$ [P$_2$W$_{17}$(FeOH$_2$)O$_{61}$] (black) and of the composites P$_2$W$_{17}$Fe@GF_N8 (red) and P$_2$W$_{17}$Fe@MWCNT_N8 (blue).

Figure 1 (black) shows the FTIR spectrum of P$_2$W$_{17}$Fe, where the typical vibrational bands of the POMs could be observed between 1100 and 700 cm$^{-1}$. The band at 1085 cm$^{-1}$ was assigned to the splitting of the $\nu_{as}$P-O stretching vibration and the three vibration bands at 947, 917 and 793 cm$^{-1}$ were assigned to the $\nu_{as}$W=O$_t$, $\nu_{as}$W-O$_b$-W and $\nu_{as}$W-O$_c$-W stretching modes (corner and edge sharing W-O-W), respectively [48,49]. The infrared spectra of the composites P$_2$W$_{17}$Fe@GF_N8 (red) and P$_2$W$_{17}$Fe@MWCNT_N8 (blue) showed weaker bands when compared with the spectrum of the POM; still, it was possible to confirm its immobilization onto the carbon materials. Additionally, bands arising from the nitrogen-doped carbon materials were also observed. After the immobilization of P$_2$W$_{17}$Fe onto the carbon materials, a small shift in the position of the characteristic bands of the POM to shorter wavelengths could be verified. The reported changes were likely due to reflect the weakening (stretching) of the W-O bonds as a result of possible changes in electron densities (or strengthening of repulsive interactions between oxygen atoms), following the adsorption of the POM on the surfaces of the carbon materials [28,50].

The XPS characterisation of the starting nitrogen-doped materials GF_N8 and MWCNT_N8 was already published by Bruno Jarrais et al. [41]. Still, for a proper comparison with the composite materials, the main peaks were recalled here. Both composites were also characterised with XPS to assess their composition. The relative amounts of the different elements were calculated through the peak areas and are presented in Table 1.

**Table 1.** XPS surface atomic percentages for the pristine carbon materials, the doped carbon materials and the prepared composites.

| Material | XPS Atomic % [a] | | | | | |
|---|---|---|---|---|---|---|
| | C1s | O1s | N1s | P2p | W4f | Fe 2p |
| GF | 96.8 | 3.2 | - | - | - | -- |
| MWCNT | 98.9 | 1.1 | - | - | - | - |
| GF_N8 | 97.0 | 2.0 | 1.0 | - | - | - |
| MWCNT_N8 | 98.0 | 1.0 | 1.0 | - | - | - |
| $P_2W_{17}Fe$@GF_N8 | 92.6 | 6.0 | 0.62 | 0.10 | 0.61 | 0.16 |
| $P_2W_{17}Fe$@MWCNT_N8 | 95.0 | 3.4 | 0.68 | 0.12 | 0.63 | 0.12 |

[a] Calculated based on the areas of the corresponding bands in the high-resolution XPS spectra.

As can be seen in Figures S2a and S3a in the Supplementary Materials file, the presence of nitrogen in the heteroatom-doped carbon materials (GF_N8 and MWCNT_N8) was found, which suggested that the doping procedure was successful. The XPS N1s spectra of the nitrogen-doped carbon materials were deconvoluted into three primary peaks, corresponding to pyridinic N at 398.2 eV, pyrrolic N at 399.6 eV and quaternary N at 401.0 eV [51,52]. For MWCNT_N8, a fourth peak at 404.1 eV was found and attributed to nitrogen oxide and/or nitrate species [51,53]. For both N-doped materials, the relative amounts of N functionalities showed the same pattern, increasing in the same order: (quaternary N) < (pyrrolic N) < (pyridinic N). The C1's high-resolution spectra (Figures S2b and S3b) were fitted with seven peaks: one at 284.6 eV, attributed to the graphitic structure ($sp^2$ and C=C), $sp^3$ hybridized carbon and C-N bonding at 285.2 eV, C-N and C-OH bonding at 285.9 eV, C-O-C groups at 286.9 eV, C=O groups (ketones, quinones and aldehydes) at 288.2 eV, COOH (carboxylic acids and esters) at 289.3 eV and a peak at 291.0 eV, attributed to $\pi$–$\pi^*$ transitions [41,54]. The O1's high-resolution spectra of the GF_N8 (Figure S2c) were fitted with one peak at 531.9 eV, attributed to C=O and COOH. However, MWCNT_N8 (Figure S3c), upon deconvolution, presented two peaks, one at 531.7 eV, assigned to C=O and COOH, and another at 533.1 eV, attributed to HO-C [38,41,55].

The XPS spectra of $P_2W_{17}Fe$@MWCNT_N8 are presented in Figure 2. The N1's XPS spectrum of the $P_2W_{17}Fe$@MWCNT_N8 composite (Figure 2a) had four main peaks at 398.5, 399.8, 401 and 402.2 eV, attributed to pyridinic, pyrrolic, quaternary N and to nitrogen oxide and/or nitrate species, respectively [41,52,56,57]. The XPS fitting of the N1's spectrum for the composite $P_2W_{17}Fe$@GF_N8 (Figure S2a) was very similar to that of GF_N8, presenting peaks at 398.1, 399.8 and 401.1 eV, corresponding to pyridinic, pyrrolic and quaternary N, respectively.

Figure 2b displayed the C1's high-resolution XPS spectrum of the $P_2W_{17}Fe$@MWCNT_N8 composite. It exhibited seven peaks in total, the primary peak at 284.6 eV ($sp^2$ graphitic structure and C=C), a peak at approximately 285.1 eV ($sp^3$ hybridized carbon and C-N bonding), a peak at 285.8 eV (C-N and C-OH bonding), a peak at approximately 286.8 eV (C-O-C groups), a peak at 288.1 eV (C=O groups, such as ketones, quinones and aldehydes), a peak at ~289.2 eV (COOH, such as carboxylic acids and esters) and a peak at 291.0 eV ($\pi$–$\pi^*$ transitions) [41,54]. The fitting of the C1's XPS spectrum of $P_2W_{17}Fe$@GF_N8 was very similar, as can be seen in Figure S4a of the Supplementary Materials file.

Figure 2c shows the high-resolution O1's spectrum for $P_2W_{17}Fe$@MWCNT_N8, which was deconvoluted into two main peaks: one at 530.7 eV, assigned to oxygen in C=O (ketones, quinones and aldehydes) and COOH (carboxylic acids and esters) groups and oxygen atoms belonging to the POM, and another at 532.6 eV, attributed to O in the HO-C groups (phenols) [38,41,55]. These two peaks were observed for MWCNT_N8 and in the O1's spectrum of $P_2W_{17}Fe$@GF_N8. It is important to refer that, after the immobilisation of the POM on the carbon materials, there was a significant increase (approximately three-fold)

in the percentage of oxygen O1s in the final composites. This was due to a contribution of the POM to the O1's percentages of the composite.

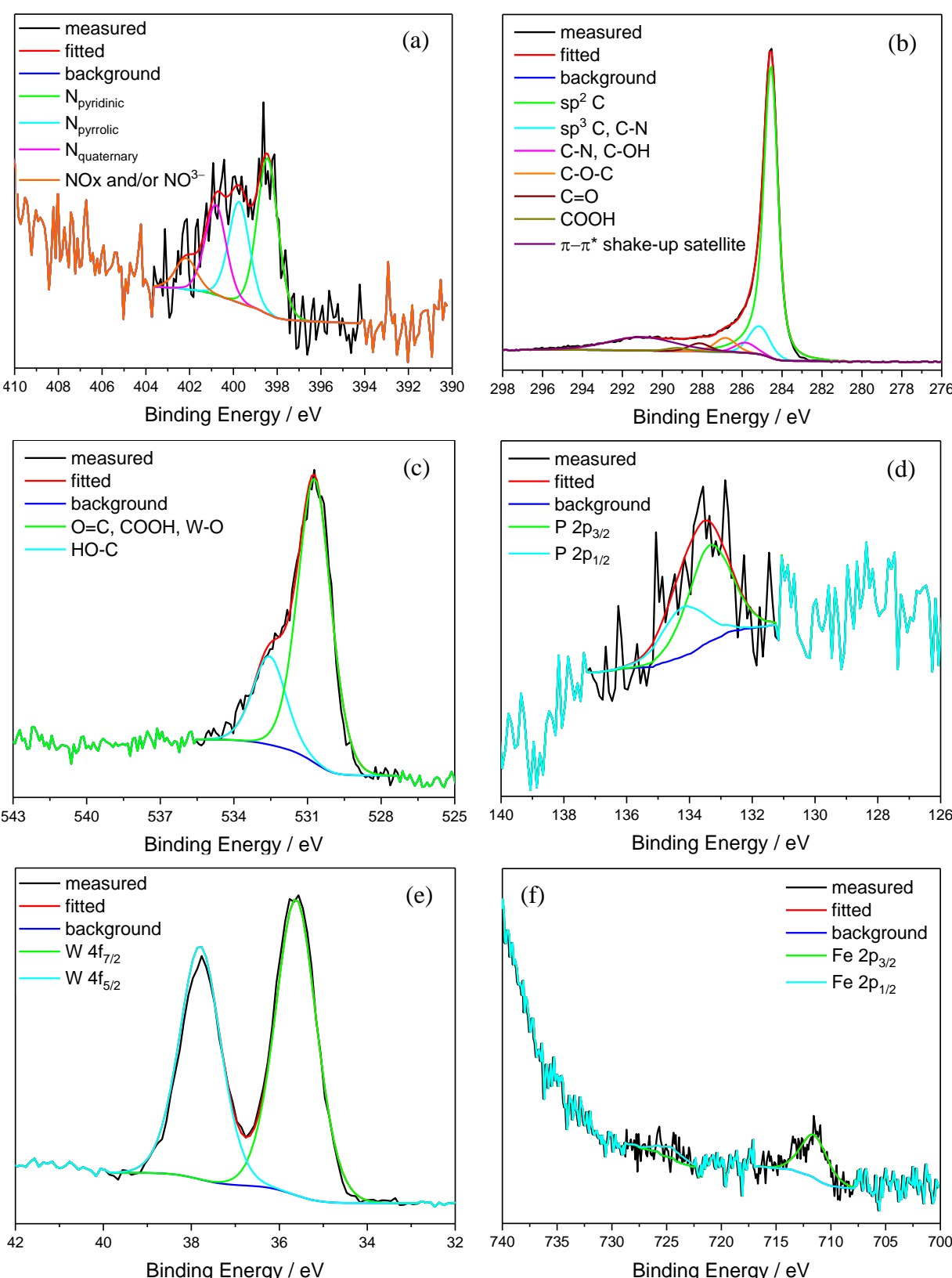

**Figure 2.** XPS deconvoluted spectra of corresponding elements in the $P_2W_{17}Fe@MWCNT\_N8$ composite.

The P2p high-resolution spectra of $P_2W_{17}Fe@CM$ are shown in Figures 2d and S4d. The observed peaks could be resolved as $2p_{3/2}$ and $2p_{1/2}$ doublets at ~133.4 and ~134.2 eV, respectively [58]. In Figures 2e and S4e, the peaks found in the high-resolution spectra of W4f were attributed to $4f_{7/2}$ (~35.6 eV) and $4f_{5/2}$ (~37.8 eV) due to spin-orbital coupling [58]. Finally, in Figure 2f and Figure S4f, the Fe2p could be divided into Fe $2p_{3/2}$ and Fe $2p_{1/2}$ with the binding energies of ~711.6 eV and ~724.7 eV associated with $Fe^{3+}$, respectively [59,60].

Assuming that XPS is a surface technique that scans sample depths down to 10 nm, this may have prevented us from reaching the expected P/W ratio. Still, the experimental value of the P/W ratio of both composites was very close to the theoretical value, thus, confirming the preservation of the polyoxometalate structure.

The morphology of all the materials underwent an assessment through scanning electron microscopy (SEM), and Figure 3 demonstrates the SEM images for $P_2W_{17}Fe@MWCNT\_N8$ (a) and $P_2W_{17}Fe@GF\_N8$ (b) at 50,000× magnification. As can be seen, the morphologies of $P_2W_{17}Fe@MWCNT\_N8$ and $P_2W_{17}Fe@GF\_N8$ differed significantly. $P_2W_{17}Fe@MWCNT\_N8$ exhibited a needle-like structure characteristic of multiwalled carbon nanotubes, whereas $P_2W_{17}Fe@GF\_N8$ showed thin and flat flakes of carbon layers. Both composites were decorated with lighter microsized spots that corresponded to $P_2W_{17}Fe$ clusters, which led to this roughened texture. In Figure 3a, the widths of the carbon nanotubes showed clear uniformity with a range of diameters between 8 and 20 nm wide. Figure 3b displays folded graphene sheets and a less rough texture. The SEM images of MWCNT_N8 and GF_N8, exhibiting the same morphological characteristics as described above, are shown in Figure S5.

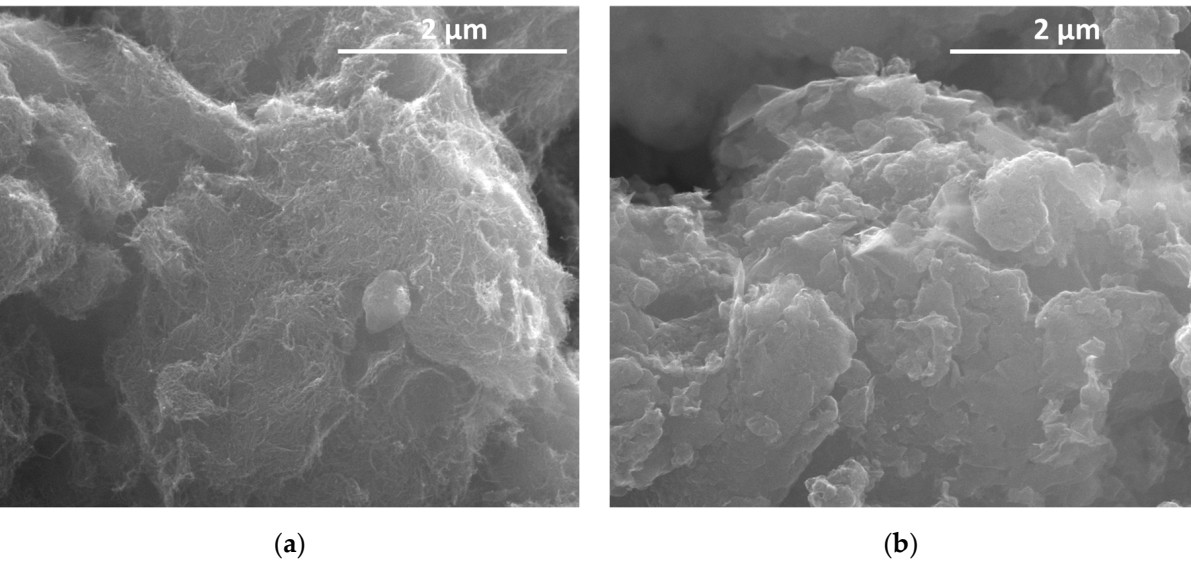

**(a)**                                    **(b)**

**Figure 3.** SEM images of $P_2W_{17}Fe@MWCNT\_N8$ (**a**) and $P_2W_{17}Fe@GF\_N8$ (**b**) at 50,000× magnification.

An EDX analysis was conducted to assess the distribution of $P_2W_{17}Fe$ throughout the composites. Figure 4 shows the SEM and EDX elemental mapping images for $P_2W_{17}Fe@MWCNT\_N8$, while the images of the $P_2W_{17}Fe$ @GF_N8 composite can be found in Figure S6. The elemental mapping analysis showed, in general, a homogenous distribution of the POM elements (W, P, Fe and O), suggesting that the POM was uniformly immobilized throughout the carbon material. The lighter shades on the EDX elemental mapping images of the W, P, Fe and O elements refer to the POM clusters.

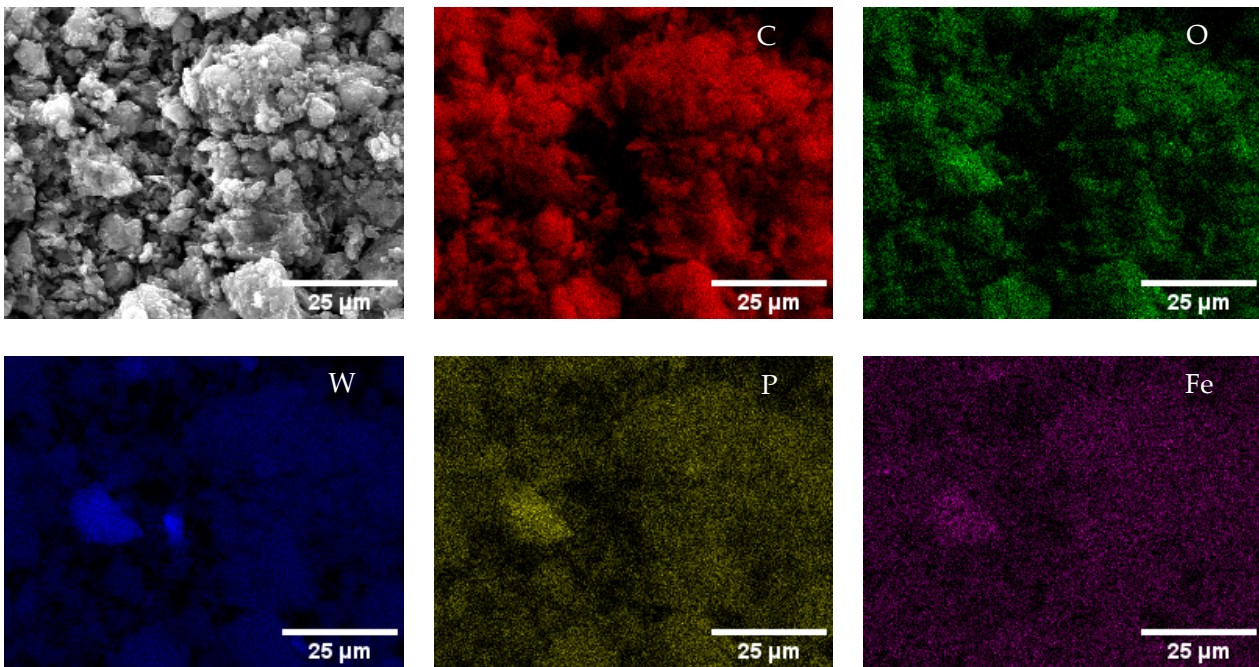

**Figure 4.** SEM and EDX elemental mapping images of $P_2W_{17}Fe@MWCNT\_N8$ at $5000\times$ magnification for the elements C (red), O (green), W (blue), P (yellow) and Fe (purple).

### 3.2. Electrochemical Performance towards the ORR

The electrocatalytic activity towards the ORR of both composites was firstly studied with cyclic voltammetry (CV) in $N_2$- and $O_2$-saturated KOH solutions (0.1 mol dm$^{-3}$). The N-doped carbon materials were also evaluated in identical experimental conditions for comparison. Figure 5 presents the CVs in $N_2$- and $O_2$-saturated solutions for the two nanocomposites prepared, where it could be clearly observed that in the $N_2$-saturated electrolyte, no peak was detected, while in the $O_2$-purged electrolyte, both composites exhibited an irreversible peak corresponding to oxygen reduction at $E_{pc}$ = 0.82 and 0.79 V vs. RHE for $P_2W_{17}Fe@MWCNT\_N8$ and $P_2W_{17}Fe@GF\_N8$, respectively. The electrochemical study of the starting nitrogen-doped materials—GF_N8 and MWCNT_N8—was published by Limani et al. [53], but was replicated here for comparison. MWCNT_N8 and GF_N8 underwent an assessment under identical experimental circumstances (refer to Figure S7a,b within the Supplementary Materials), and they exhibited reduction peaks at $E_{pc}$ = 0.74 and 0.80 V vs. RHE, respectively. The performance of Pt/C (20 wt %) was also tested under the same experimental conditions (Figure S7c), exhibiting the ORR peak at $E_{pc}$ = 0.86 V vs. RHE.

To gain a deeper understanding of the electrocatalytic performance of the $P_2W_{17}Fe@CM$ nanocomposites towards the ORR, linear sweep voltammetry (LSV) was carried out at 1600 rpm in an $O_2$-saturated KOH solution (0.1 mol dm$^{-3}$). The LSV curves shown in Figure 6a were obtained by subtracting the blanks (the latter recorded in a $N_2$-saturated KOH solution) from the ones saved in the $O_2$-saturated KOH. Table 2 displays the key ORR parameters acquired—the onset potential ($E_{onset}$), the diffusion-limiting current density values ($j_{L, 0.26 V, 1600 rpm}$), the Tafel slopes (TS) and the number of electrons transferred per oxygen molecule ($n_{O2}$).

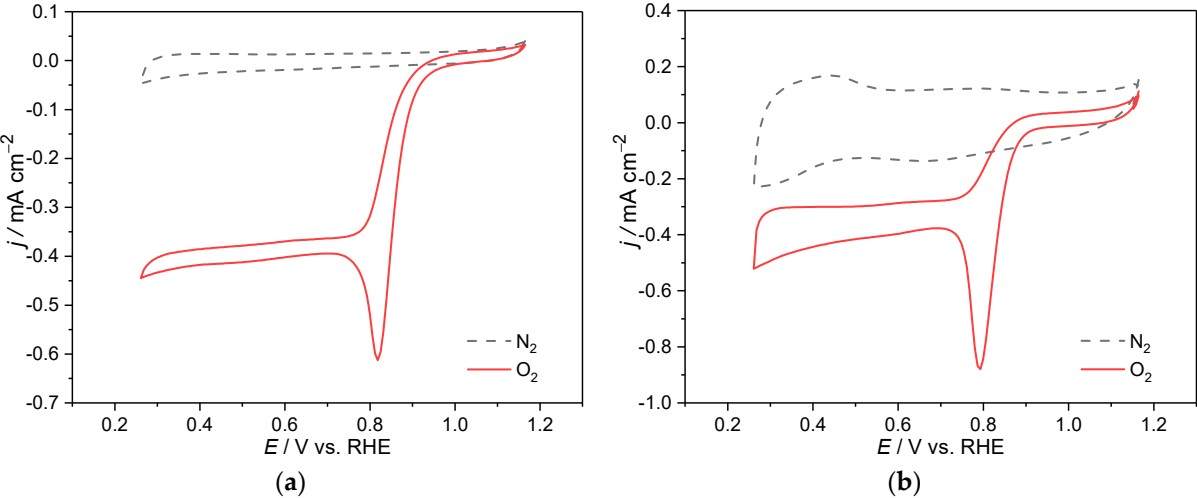

**Figure 5.** CVs of $P_2W_{17}Fe@MWCNT\_N8$ (**a**) and $P_2W_{17}Fe@GF\_N8$ (**b**) modified electrodes in $N_2$-saturated (dash line) and $O_2$-saturated (full line) 0.1 mol dm$^{-3}$ KOH solution at 0.005 V s$^{-1}$.

**Table 2.** Main ORR parameters obtained from polarisation curves recorded in $O_2$-saturated 0.1 mol dm$^{-3}$ KOH solution for the different materials tested.

| Samples | $E_{onset}$ vs. RHE (V) | $j_L$ (mA cm$^{-2}$) | Tafel Slope (mV dec$^{-1}$) | $n_{O2}$ |
|---|---|---|---|---|
| Pt/C | 0.94 | −4.7 | 89 | 4.0 |
| GF | 0.72 | −2.0 | 63 | 2.4 |
| MWCNT | 0.69 | −2.0 | 54 | 2.1 |
| GF_N8 | 0.83 | −3.3 | 70 | 2.5 |
| MWCNT_N8 | 0.83 | −4.0 | 47 | 3.5 |
| $P_2W_{17}Fe@GF\_N8$ | 0.83 | −3.3 | 58 | 3.1 |
| $P_2W_{17}Fe@MWCNT\_N8$ | 0.84 | −3.9 | 48 | 3.4 |

The $E_{onset}$ values of the $P_2W_{17}Fe@CM$ nanocomposites and N-doped carbon materials were very similar, ranging from 0.83 to 0.84 V vs. RHE. $P_2W_{17}Fe@MWCNT\_N8$ had the closest $E_{onset}$ value to the one obtained for Pt/C (0.94 V vs. RHE). Furthermore, all electrocatalysts showed $j_L$ values between −4.0 and −3.3 mA cm$^{-2}$, but did not reach the value obtained for Pt/C (−4.7 mA cm$^{-2}$). Still, these nanocomposites performed well compared to other similar electrocatalysts, as can be observed in Table S2. It is also important to highlight that the ORR electrocatalytic activity of the pristine carbon materials greatly improved after nitrogen doping, which agreed with various studies that reported an enhancement in the ORR electrocatalytic performance after the N-doping of similar carbon materials [53,61]. After N-doping, the $E_{onset}$ potential of MWCNT shifted by 14 mV towards positive values, and the value of $j_L$ doubled, whereas GF presented a positive shift of 11 mV and a $j_L$ 1.3 times higher (in magnitude of $j_L$). As nitrogen atoms have a high electronegativity, there may have contributed to an increased favourable interaction between $O_2$ molecules promoting the ORR mechanism through a rearrangement of the electronic structure of the neighbouring atoms [62].

Based on the XPS results in Table 1, it was not possible to directly link the superior performance of MWCNT_N8 with N%, as both the nitrogen-doped carbon materials had the same total percentage. Therefore, Table S1 presents the relative atomic percentages of nitrogen in the prepared carbon materials. According to the literature, the most active N species for the ORR are the pyridinic and quaternary ones. Pyridinic nitrogen atoms, due to their electron donation to the conjugated p bond of graphene and to the lone pair of electrons, have been identified as electroactive sites for the ORR through their ability to facilitate reductive oxygen adsorption [62,63]. Similarly, quaternary nitrogen is also recognised as an electroactive site for the ORR [63,64]. Even though our previous work [58]

showed that the amount of pyridinic nitrogen is positively correlated with the ORR activity in nitrogen-doped carbon materials, here, it seems that quaternary nitrogen was more relevant. If we compare MWCNT_N8 and GF_N8, the best performing electrocatalysts was the former presenting higher quaternary N (16.9%). The same applied for the composite material.

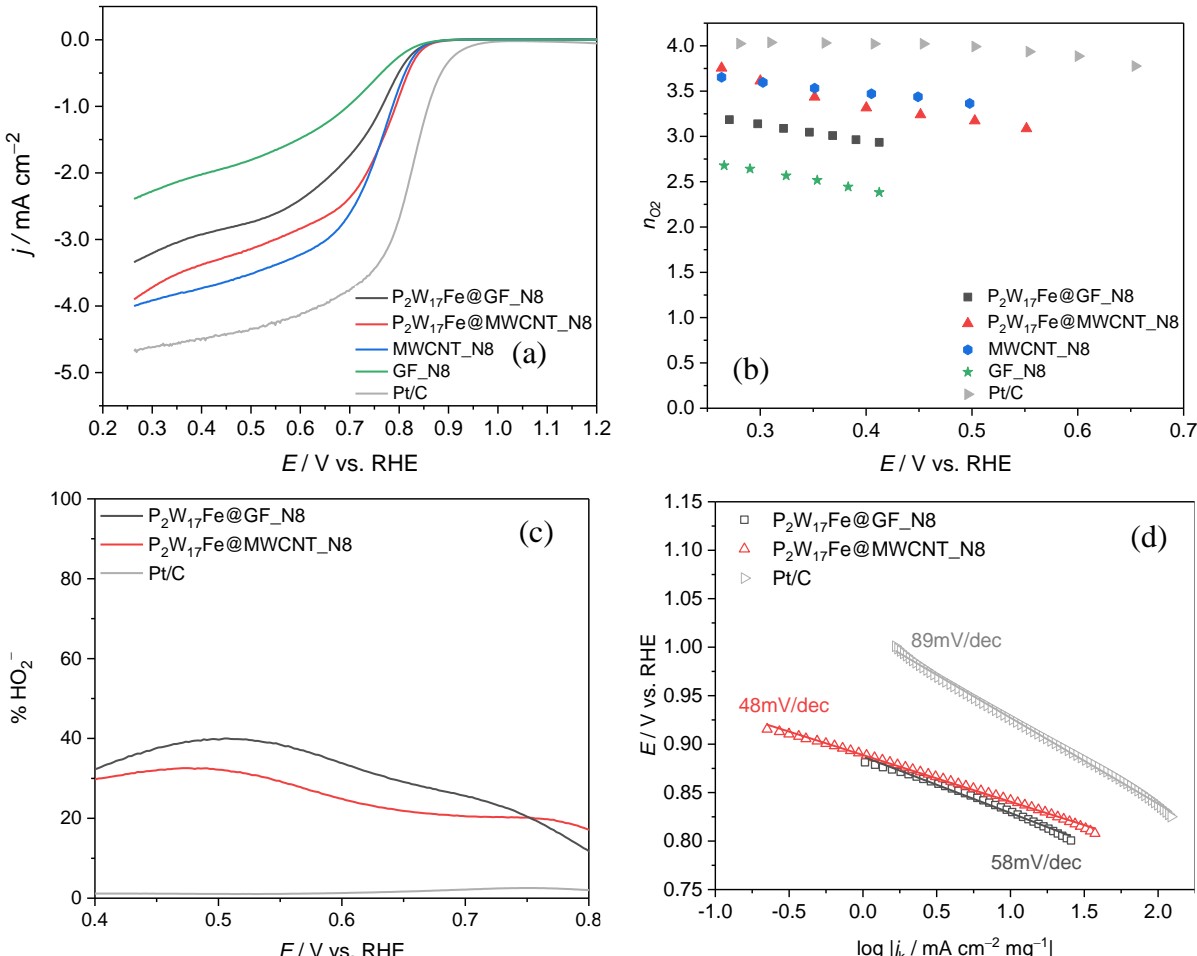

**Figure 6.** ORR LSV curves acquired using an RDE in $O_2$-saturated KOH solution (0.1 mol dm$^{-3}$) at 1600 rpm and 0.005 V s$^{-1}$ of commercial Pt/C (20 wt %), $P_2W_{17}Fe@CM$ nanocomposites and nitrogen-doped carbon materials (**a**); $n_{O2}$ at several potential values (**b**); estimated percentage of $HO_2^-$ produced with RRDE for $P_2W_{17}Fe@CM$ and Pt/C (20 wt %) in $O_2$-saturated 0.1 mol dm$^{-3}$ KOH solution at 1600 rpm; the disk potential was swept at v = 0.005 V s$^{-1}$ and the ring potential was kept at E = 0.2 V versus Ag/AgCl (**c**); ORR Tafel plots for $P_2W_{17}Fe@CM$ obtained from LSV data in Figure 6a, where current densities were normalized to the mass of each EC (**d**).

On the other hand, we had the incorporation of POMs, which are known for their excellent ORR activity. POMs have multiple redox-active sites that can facilitate an electron transfer during the ORR. In previous studies, including ours, POMs have been suggested to interact with nitrogen-doped carbon materials to form composite catalysts that exhibit enhanced ORR activity.

Figure 6b depicts the quantity of electrons transferred per oxygen molecule for the N-doped CM and the respective composites. The results obtained with $P_2W_{17}Fe@MWCNT_N8$ and MWCNT_N8 suggested that the redox processes were very close to being four-electron ones. On the other hand, $P_2W_{17}Fe@GF_N8$ and GF_N8 participated in a mixed redox process involving two and four electrons, as confirmed with the $n_{O2}$ values. Based on these parameters, the behaviour after POM immobilisation appeared to be very similar to the

nitrogen-doped carbon materials. A possible explanation is that the active sites originating from the POM immobilisation did not offset for the eventual loss of N-induced active sites [38,53].

In an alkaline medium, the oxygen reduction reaction can take either an indirect (two-electron) pathway, which involves reducing $O_2$ to $HO_2^-$ and then reducing the intermediate to $H_2O/HO^-$, or a more desirable process, where $O_2$ is directly reduced to $H_2O/HO^-$ [65]. Thus, for a more complete examination of the ECs, it is of utmost importance to estimate the percentage of $HO_2^-{}_2$ produced during the ORR process. The percentage of $HO_2^-$ produced was calculated using Equation (S4) (Supplementary Materials) to various potential values (Figure 6c). The $P_2W_{17}Fe@MWCNT\_N8$ composite showed the lowest percentage of $HO_2^-$ produced (33%), which corresponded to the $n_{O2}$ value ($n_{O2} = 3.4$). For the $P_2W_{17}Fe@GF\_N8$ composite, the values of $HO_2^-$ produced (40%) were also in accordance with the $n_{O2}$ value obtained (3.1). Despite the agreement of the results, care should be taken when a direct comparison between these two methods is attempted, as both present limitations [61,66–68]. Firstly, the oxidation process of hydrogen peroxide on the platinum electrocatalyst was not a mass-transfer-limited process. Secondly, as can be seen in the SEM images, these prepared electrocatalysts presented a rough structure of the support material and a heterogeneity of clusters of the immobilised POM. This may have altered the electrode geometry and introduced turbulence in the electrolyte flow, resulting in approximate $n_{O2}$ measurements that may not accurately portray the true electrocatalytic performance of the ORR [53].

In order to obtain more information about the kinetics of the reaction, we extracted the Tafel plots for the prepared nanocomposites and the Pt/C reference from the LSV curves in $O_2$-saturated KOH at 1600 rpm. These plots are presented in Figure 6d. The nanocomposite $P_2W_{17}Fe@GF\_N8$ had a slightly higher Tafel slope value (58 mV dec$^{-1}$) when compared with $P_2W_{17}Fe@MWCNT\_N8$ (48 mV dec$^{-1}$), but both nanocomposites showed a lower Tafel slope compared with Pt/C (89 mV dec$^{-1}$). These results indicated that oxygen molecules were readily adsorbed and activated on the surface of these materials. In general, the ORR process can follow different mechanisms in a high-pH environment. One is the well-known electrocatalytic inner-sphere electron transfer mechanism, where molecular $O_2$ undergoes direct chemisorption on the catalyst site, leading to a direct/series 4e$^-$ pathway without the desorption of reaction intermediates (such as peroxide) from the surface [69–71]. Normally, metal-containing materials follow a dissociative four-electron pathway, facilitated by their excellent $O_2$ adsorption ability. This process involves the initial adsorption of $O_2$, followed by breakage of the O-O bond, forming adsorbed O* species ($O_2 + 2^* \rightarrow 2O^*$, where * denotes a surface catalytic site). The species then gains two electrons and two protons to produce the final product OH- ($2O^* + 2e^- + 2H_2O \rightarrow 2OH^* + 2OH^-$; $2OH^* + 2e^- \rightarrow 2OH^- + 2^*$) [72].

Our Tafel slope values suggested that for $P_2W_{17}Fe@CM$, the global reaction rate was ruled by the conversion of a species $MOO^-$ to MOOH (where M denotes an empty site on the electrocatalyst surface) or by the first discharge step [38,53].

The proportional relationship between the double-layer capacitance ($C_{dl}$) and the electrochemically active surface areas (ECSAs), as well as the similarity between the electrocatalysts used in this study, allowed for a direct comparison of the obtained $C_{dl}$ values of the composites [73–76]. By conducting CV measurements at varying scan rates, it was possible to estimate $C_{dl}$ values from the slopes of the linear fittings of the CV current densities (measured at the same potential of 1.13 V vs. RHE in a nonfaradaic region), obtained at different scan rates for the composites and their nitrogen-doped carbon materials (see experimental details in Supplementary Materials file Section S1.4 and Figures S13 and S14). Still, the $C_{dl}$ should be interpreted as an estimation of the number of accessible electrocatalytically active sites for a specific electrocatalyst [77]. Table 3 clearly shows a notable increase in the count of active sites, following the immobilisation of the POM ($P_2W_{17}Fe$) on both nitrogen-doped carbon materials. This increase was higher for the $P_2W_{17}Fe@MWCNT\_N8$ composite, with a composite/carbon support $C_{dl}$ ratio of 3.98 vs. 1.39 for $P_2W_{17}Fe@GF\_N8$.

**Table 3.** Electrocatalytically active surface area (ECSA) values of the carbon supports and their corresponding POM-based composites.

| CM Support | $C_{dl}$ (mF cm$^{-2}$) | Composite | $C_{dl}$ (mF cm$^{-2}$) | Composite/Carbon Support $C_{dl}$ Ratio |
|---|---|---|---|---|
| GF_N8 | 0.0228 | $P_2W_{17}Fe@GF\_N8$ | 0.0317 | 1.39 |
| MWCNT_N8 | 0.0203 | $P_2W_{17}Fe@MWCNT\_N8$ | 0.0808 | 3.98 |

When evaluating the potential of an electrocatalyst for the ORR, it is of utmost importance to assess its stability. This was carried out with chronoamperometry (CA) at $E$ = 0.46 V vs. RHE for 36,000 s at 1600 rpm in an oxygen-saturated alkaline electrolyte (Figure 7a). The nanocomposite $P_2W_{17}Fe@MWCNT\_N8$ showed superior stability with an 88% current retention, surpassing the results achieved for Pt/C (86%), while $P_2W_{17}Fe@GF\_N8$ showed a lower current retention value (68%). The increased current retention observed for $P_2W_{17}Fe@MWCNT\_N8$ could be attributed to the stronger interaction between the POM and its highly ordered nitrogen-doped MWCNT. Still, the good stability of the composites was mainly due to the trapping of the POM particles in the nanonetworks of the carbon materials, avoiding (or at least limiting) their migration, aggregation and leaching towards the electrolyte, not adversely affecting the corresponding support stabilities. In addition, the high degree of graphitisation and/or ordered wall structure of the carbon materials could reduce a possible degradation of the composite, which could be another cause of the increased durability [78–80].

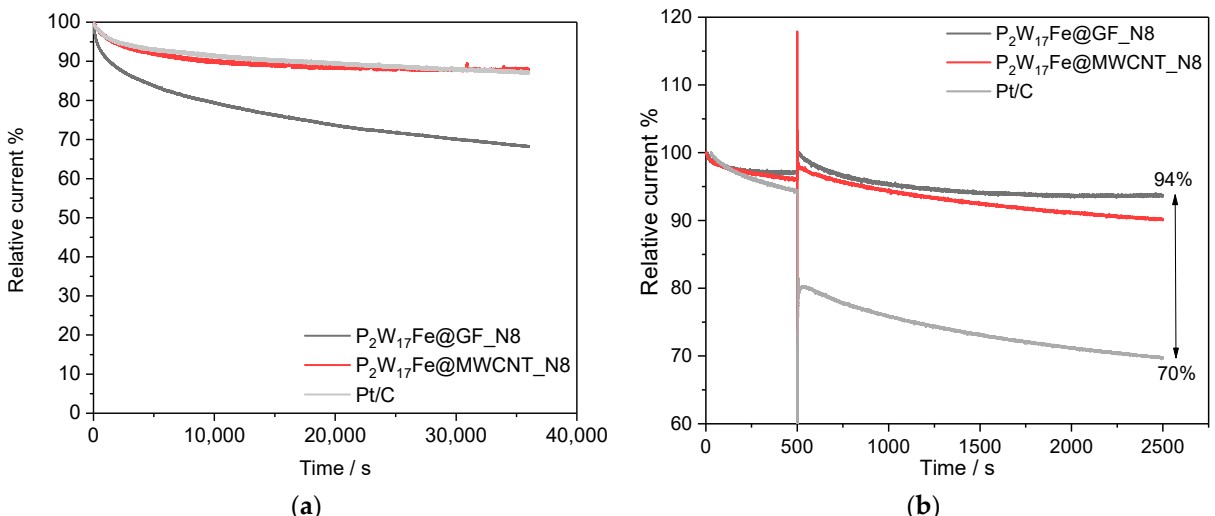

**(a)**　　　　　　　　　　　　　　　　　　　　**(b)**

**Figure 7.** Chronoamperometric response of the ECs at 1600 rpm in 0.1 mol dm$^{-3}$ O$_2$-saturated KOH at E = 0.46 V vs. RHE for 36,000 s (**a**) and E = 0.46 V vs. RHE with the addition of 0.5 mol dm$^{-3}$ methanol after $\approx$ 500 s (**b**).

Methanol is an excellent hydrogen carrier fuel, and can also be produced from sustainable and green pathways to allow it to be a carrier of low carbon and potentially carbon-neutral hydrogen [81,82]. As such, the study of the methanol tolerance of electrocatalysts in methanol fuel cells is extremely important, since the methanol crossover from the anode to the cathode can lead to a significantly decrease in the cathodic performance, if the electrocatalysts is sensitive to it [83]. The Pt-based materials are handicapped by their high sensitivity to the methanol oxidation reaction, despite their better performance in alkaline medium compared most of the studied electrocatalysts [84]. This situation affected their performance in the ORR, lowering the current density obtained. Thus, the tolerance of methanol was assessed through chronoamperometry in O$_2$-saturated 0.1 mol dm$^{-3}$ KOH in the presence of methanol (0.5 mol dm$^{-3}$) for the Pt/C and $P_2W_{17}Fe@CM$ composites during

2500 s (Figure 7b). Methanol was added 500 s after the beginning of the CA. The Pt/C electrocatalyst maintained only ~70% of its initial current density, whereas the composites retained between ~90 and 94%, being one of the advantages of using this type of material.

## 4. Conclusions

In this work, it was possible to successfully immobilise the Wells–Dawson POM $[P_2W_{17}(FeOH_2)O_{61}]^{7-}$ onto graphene flakes and carbon nanotubes doped with nitrogen ($P_2W_{17}Fe@GF\_N8$ and $P_2W_{17}Fe@MWCNT\_N8$) without the need for linker molecules. It was proven that both composites were successfully synthesised using several characterisation techniques, such as FTIR, XPS and SEM analysis. The prepared composites showed intrinsic electroactivity towards the ORR in an alkaline medium, displaying onset potential values varying between 0.83 and 0.84 V vs. RHE. Furthermore, the nanocomposites demonstrated low Tafel slopes (47–70 mV dec$^{-1}$) and commendable $j_L$ values ($-3.3$–$-4.0$ mA cm$^{-2}$). The analysis of the Koutecky−Levich plots revealed that the ORR process occurred through a mixed electron process for $P_2W_{17}Fe@GF\_N8$ and a process involving four electrons for $P_2W_{17}Fe@MWCNT\_N8$, also confirmed with the results of the %$HO_2^-$ production. The composite $P_2W_{17}Fe@MWCNT\_N8$ demonstrated a sturdy electrocatalytic performance over 36,000 s, revealing a high retention value of 88%—surpassing the commercial Pt/C, which achieved 86%. Additionally, both composites had an excellent methanol tolerance (90–94%), evincing high selectivity for the OR and, thus, overcoming a significant limitation of the Pt/C electrocatalyst. This study was a significant stride towards creating new, inexpensive and sturdy electrocatalysts that are free of noble metals, with immense potential for energy-related reactions.

**Supplementary Materials:** The following supporting information can be downloaded at: https://www.mdpi.com/article/10.3390/inorganics11100388/s1, Figure S1: FTIR spectra in the 4000–500 cm$^{-1}$ range of the nitrogen-doped carbon materials GF_N8 (black) and MWCNT_N8 (red), Figure S2: XPS deconvoluted spectra of corresponding elements of GF_N8, Figure S3: XPS deconvoluted spectra of corresponding elements of MWCNT_N8, Figure S4: XPS deconvoluted spectra of corresponding elements in the $P_2W_{17}Fe@GF\_N8$ composite, Figure S5: SEM images of MWCNT_N8 (a) and GF_N8 (b) at 50,000× magnification, Figure S6: SEM and EDX elemental mapping images of $P_2W_{17}Fe$ @GF_N8 at 5000× magnification for the elements C (red), O (green), W (blue), P (yellow) and Fe (purple), Figure S7: CVs of MWCNT_N8 (a), GF_N8 (b) and Pt/C (20 wt. %) (c) modified electrodes in N$_2$-saturated (dash line) and O$_2$-saturated (full line) 0.1 mol dm$^{-3}$ KOH solution at 0.005 V, Figure S8: ORR LSV polarization curves for Pt/C (20 wt. %) at different rotation rates in O$_2$-saturated 0.1 mol dm$^{-3}$ KOH solution at 0.005 V s$^{-1}$ (a), and the corresponding Koutecky–Levich (K-L) plots (b), Figure S9: ORR LSV polarization curves for MWCNT_N8 at different rotation rates in O$_2$-saturated 0.1 mol dm$^{-3}$ KOH solution at 0.005 V s$^{-1}$ (a) and the corresponding Koutecky–Levich (K-L) plots (b), Figure S10: ORR polarization curves for GF_N8 at different rotation rates in O$_2$-saturated 0.1 mol dm$^{-3}$ KOH solution at 0.005 V s$^{-1}$ (a) and the corresponding Koutecky–Levich (K-L) plots (b), Figure S11: ORR LSV polarization curves for $P_2W_{17}Fe@MWCNT\_N8$ at different rotation rates in O$_2$-saturated 0.1 mol dm$^{-3}$ KOH solution at 0.005 V s$^{-1}$ (a) and the corresponding Koutecky–Levich (K-L) plots (b), Figure S12: ORR LSV polarization curves for $P_2W_{17}Fe@GF\_N8$ at different rotation rates in O$_2$-saturated 0.1 mol dm$^{-3}$ KOH solution at 0.005 V s$^{-1}$ (a) and the corresponding Koutecky–Levich (K-L) plots (b), Figure S13: CVs at different scan rates for GF_N8 (a), MWCNT_N8 (b), P2W17Fe@GF_N8 (c), $P_2W_{17}Fe@MWCNT\_N8$ (d) and Pt/C (e) in N$_2$-saturated KOH (0.1 M), Figure S14: current density-scan rate linear fitting plots for all materials. Numeric values correspond to the double-layer capacitances ($C_{dl}$) for each material, Table S1: relative atomic percentages of nitrogen presented in the XPS high-resolution N1s spectra of the prepared carbon materials, Table S2: ORR performance of transition metal carbon hybrid electrocatalysts obtained from the literature. References [38,45,48,53,58,85–97] cited in Supplementary Materials.

**Author Contributions:** Conceptualization, D.M.F. and P.d.O.; methodology, H.C.N., B.J., I.-M.M. and A.-L.T.; validation, D.M.F.; writing—original draft preparation, H.C.N.; writing—review and editing, D.M.F. and P.d.O.; supervision, D.M.F. and C.F.; funding acquisition, C.F. All authors have read and agreed to the published version of the manuscript.

**Funding:** This work was financially supported by project UNIRCELL POCI-01-0145-FEDER-16422 and funded by FEEI through Programa operacional Competitividade e Internacionalização-COMPETE2020 and by national funds through FCT—Fundação para a Ciência e a Tecnologia, I.P.

**Data Availability Statement:** The data presented in this study are available on request from the corresponding author. The data are not publicly available due to institutions policies.

**Acknowledgments:** Thanks are due to the PT national funds FCT/MCTES through the projects UIDB/50006/2020 and UIDP/50006/2020, as well as to Campus France—project PHC Pessoa 42341RM (O2@nanoCPOM). Thanks are also due to the FCT project FOAM4ENER PTDC/QUI-ELT/28299/2017. H.C.N. thanks the FCT through the ChemMat program for his PhD grant PD/BD/143095/2018. D.M.F. thanks FCT/MCTES for funding through the Individual Call to Scientific Employment Stimulus (2021.00771.CEECIND/CP1662/CT0007). I.-M M., A.-L.T. and P.d.O. thank the CNRS (namely, its IEA programme) and the Université Paris-Saclay for financial support.

**Conflicts of Interest:** The authors declare that they have no conflict of interest.

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
