# Peer review of "Hybrids Composed of an Fe-Containing Wells–Dawson Polyoxometalate and Carbon Nanomaterials as Promising Electrocatalysts for the Oxygen Reduction Reaction"

_inorganics, doi:10.3390/inorganics11100388_

Round 1

Reviewer 1 Report

In this manuscript, two catalysts P2W17Fe@GF_N8 and P2W17Fe@MWCNT_N8 were prepared by immobilizing Wells-Dawson POM salt K7[P2W17(FeOH2) O61] ·20H2O on graphene sheets and nitrogen-doped multi-walled carbon nanotubes. The two catalysts were characterized by FTIR, XPS and SEM to prove the successful preparation of the composites. The ORR performance and long-term electrochemical stability of these two materials were tested in alkaline electrolyte. The data are relatively comprehensive and sufficiently novel, which is important for this study and can be published with some minor modifications.

1. As far as we know, there will be a slight shift in the peak of XPS test for different samples. In Fig.3b, it is observed that the author deconvoluted the spectrum of C into seven peaks. What is the basis for dividing into seven peaks? Can it be divided into 6 peaks?

2. What is the basis for calculating the electron transfer number and HO2-yield in Figs 6b and 6c? Why is the graph of electron transfer number a scatter diagram rather than a curve?

3.What is the product of 2 electrons under alkaline conditions? The ordinate of Fig.6c should be “HO2- %” instead of “% H2O2”. Please also check your description in the comments below the article and the figure.

4. The author should perform a K-L analysis, obtain the value of “n” from the K-L diagram, and then check whether it matches the value already obtained from the formula given in the manuscript.

5.There are some format problems in the manuscript. The correct use of spaces is also recommended to pay more attention to, for example, between numbers and units to have spaces. Therefore, it is recommended to conduct a comprehensive revision and inspection of the full text.

6.The water in the Abstract K7[P2W17(FeOH2) O61].20H2O is bound water, and the writing method should be corrected.

Reviewer 2 Report

This manuscript describe thepreperation of carbon nanomaterilal with POM and their EC performance. My recomendation to the authors are as followed.

1. Quantitativity with XPS is not reliable. carry out the other normal elemental analyses, such as CHN AAS and ICP.

2. I did not follow the meaning of the utilization of Fe-substituted POM. Compare the EC data with fully occupied P2W18, Fe sources, the physical mixture of P2W18 and the physical mixture of Na2WO4 and Fe sources. It is not difficult to add these data, and it makes this manusctipt better.

3. I cannot find "Figure 2" in the main text. Add it at the appropriate possition.
